# RE-READING IMPROVES REASONING IN LANGUAGE MODELS

## ABSTRACT

Reasoning presents a significant and challenging issue for Large Language Models (LLMs). The predominant focus of research has revolved around developing diverse prompting strategies to guide and structure the reasoning processes of LLMs. However, these approaches based on decoder-only causal language models often operate the input question in a single forward pass, potentially missing the rich, back-and-forth interactions inherent in human reasoning. Scant attention has been paid to a critical dimension, i.e., the input question itself embedded within the prompts. In response, we introduce a seemingly straightforward yet remarkably effective prompting strategy—RE2, which involves re-reading the question. Drawing inspiration from human learning and problem-solving, re-reading entails revisiting the question information embedded within input prompts. This approach aligns seamlessly with the cognitive principle of reinforcement, enabling LLMs to understand the input in a "bidirectional" manner, extract deeper insights, and ultimately enhance their reasoning capabilities across various tasks. Experiments conducted on a series of reasoning benchmarks serve to underscore the effectiveness and generality of our method. Moreover, our findings demonstrate that our approach seamlessly integrates with various language models, though-eliciting prompting methods, and ensemble techniques, further underscoring its versatility and compatibility in the realm of LLMs.

## 1 INTRODUCTION

In the ever-evolving landscape of artificial intelligence, large language models (LLMs) have emerged as a cornerstone of natural language understanding and generation Brown et al. (2020); Touvron et al. (2023a;a); OpenAI (2023). However, as these models have grown in size and complexity, a pivotal challenge has come to the forefront: imbuing them with the ability to reason effectively. The capacity to engage in sound reasoning is a hallmark of human intelligence, enabling us to infer, deduce, and solve problems. In LLMs, this skill is paramount for enhancing their practical utility across a multitude of tasks. Despite their remarkable capabilities, LLMs often struggle with nuanced reasoning Blair-Stanek et al. (2023); Arkoudas (2023), prompting researchers to explore innovative strategies to bolster their reasoning prowess Wei et al. (2022b); Gao et al. (2023); Besta et al. (2023).

The existing body of research in this domain has predominantly concentrated on designing diverse thought-eliciting prompting strategies to guide and channel their reasoning processes. Noteworthy strategies such as Chain-of-Thought (CoT) Wei et al. (2022b), Tree of Thoughts (ToT) Yao et al. (2023a), Graph of Thoughts Besta et al. (2023), Plan-and-Solve (PS) Wang et al. (2023a), and program-aided language model (PAL) Gao et al. (2023) have emerged to structure and elicit logical trains of thought from these models. However, existing decoder-only causal language modeling (CLM) architecture often operates in a single forward pass, which may miss the richer, back-and-forth interactions that humans use when reasoning through a challenging problem. Meanwhile, an intriguing observation emerges – while significant efforts have been directed towards molding the path of reasoning, CoT family for example, scant attention has been paid to a critical dimension, i.e., the input problem itself embedded within the prompts.

Herein lies the foundation of our approach, wherein we present a deceptively simple yet profoundly effective prompting strategy: *re-reading*, or RE2 for short, which involves reading the question twice (see Figure 1). Drawing inspiration from human learning and problem-solving processes, we posit

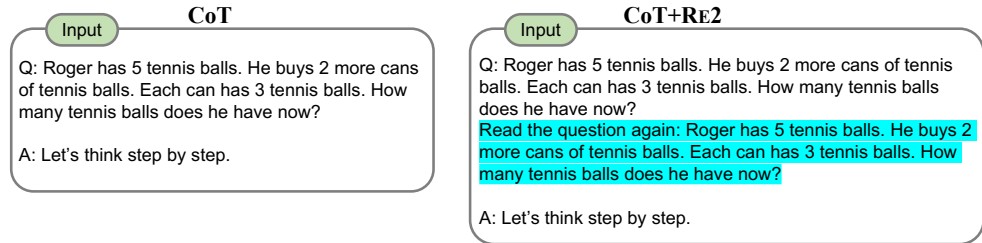

Figure 1: Example inputs of CoT prompting and CoT prompting with our RE2 method.

that revisiting the question information embedded in the input prompts can reassess the context, refine understanding, and correct potential misconceptions. This strategy aligns with the cognitive principle of reinforcement, allowing the models to iteratively build upon their initial understanding of the problem. By engaging in multiple passes over the input, the models can glean deeper insights, identify intricate patterns, and construct more nuanced connections that contribute to heightened reasoning outcomes. From the perspective of LLMs' architecture, RE2 facilitates "bidirectional" comprehension of questions within the context of decoder-only causal language models. Our re-reading mechanism is far simpler than existing approaches that either perform the reasoning with multiple stages of prompting Zhou et al. (2022) or sample multiple reasoning paths Wang et al. (2023c) to improve generation quality. Besides, our re-reading works off-the-shelf with various pre-trained language models and prompting strategies as a "plug & play" module, and avoids any intricate stages of prompting or sampling.

To validate our re-reading strategy's efficacy, we conducted extensive experiments spanning arithmetic, commonsense, and symbolic reasoning tasks. Our findings consistently show significant improvements in reasoning performance, especially when using the chain-of-thought prompting method. Moreover, our model exhibits versatility across various language models, task settings, and prompting methods, highlighting its broad applicability.

## 2    RELATED WORK

**Reasoning with Large Language Models.**    LLMs represent a significant milestone in the journey towards artificial general intelligence (AGI) OpenAI (2023); Touvron et al. (2023b) . Their remarkable abilities span a broad range of tasks, facilitated through a unified natural language interface that operates in a generative manner. Here, reasoning ability is particularly crucial on the way towards AGI, where artificial intelligence needs to act or think like human beings Qiao et al. (2023); Huang & Chang (2023). In the literature on LLMs, performing reasoning tasks via interaction in natural language plays a significant role in evaluating an LLM, into which academia and industry have been dedicating many endeavors Wei et al. (2022a); Suzgun et al. (2023a); Turpin et al. (2023). In principle, most works for reasoning with large language models could fall into the paradigm of "Chain-of-Though" Wei et al. (2022b); Kojima et al. (2022), which assists LLMs in fulfilling complex reasoning tasks by generating intermediate steps explicitly. Therefore, most of the endeavors are dedicated to improving the basic principle by the following aspects: i) the structure of "chain", e.g., tree Yao et al. (2023a), graph Yao et al. (2023b); ii) the modality of the chain, e.g., program Gao et al. (2023); iii) the reliability of the chain, e.g., self-consistency Wang et al. (2023c), faithful Lyu et al. (2023), retrieval-based verifying He et al. (2023); and iv) decomposition of the chain, e.g., least-to-most Zhou et al. (2023), decomposed Radhakrishnan et al. (2023), plan-to-solve Wang et al. (2023a). In contrast, our simple re-reading strategy for LLMs is orthogonal to these improvements via a trade-off between the intermediate steps and the query itself. Besides, our re-reading strategy is complementary to many previous works by preventing the answer from being derived overwhelmingly from the CoT but overlooking the original query.

**Re-reading Strategy in Text Understanding.**    In deep learning, the success of performing text-understanding tasks Song et al. (2018); Luo et al. (2019a); Yang et al. (2019); Lei et al. (2019) depends on the heuristics of human reading strategy, e.g., pre-reading, reading and post-reading Saricoban (2002); Toprak & ALMACIOĞLU (2009); Pressley & Afflerbach (2012). Specifically, many effective algorithms have been crafted around the idea of re-reading. Although deep architectures, from multi-layer Bi-LSTM Huang et al. (2015) to Transformer-encoder Vaswani et al. (2017), have their mechanisms that provide a form of "re-reading", the notion that simply processing an

input once might not be sufficient for understanding or generating a complex output has been long-standing. Initially, Sha et al. (2016) and Sha et al. (2017) found that repeated reading mechanisms do improve performance on some tasks, e.g., sentiment analysis, semantic relation classification, and event extraction. Then, Liu & Li (2016) propose to mimic the repeated reading strategy and present neural networks with multi-level attention, which is proven effective in recognizing implicit discourse relations. Sequentially, Zhu et al. (2018) propose a multi-glance mechanism, modeling the habit of reading behavior, which can benefit a wide range of tasks. More recently, Luo et al. (2019b) adopt a network to encode the gist of paragraphs for rough reading and a decision-making policy for careful reading, which can improve extractive summarization. Therefore, it is natural to introduce a re-reading strategy to large language models since i) the Transformer-decoder architecture of LLMs, with mono-directional attention mechanisms, hinders the implicit re-reading capability, and ii) the context combined with the input query to prompt LLMs could be intricate, including streamed text, background information, external knowledge, intermediate rationale, and few-shot demonstrations, thus overwhelming the original target query.

**Knowledge Recall.** From the perspective of information seeking, prompting LLMs can be seen as a sort of "knowledge recall" via a parametric fashion, where the prompt can be seen as a retrieval query. In contrast to conventional non-parametric retrieval – vector database Karpukhin et al. (2020); Izacard et al. (2022) for example, the LLM as a neural knowledge model Bosselut et al. (2019); AlKhamissi et al. (2022) can easily generalize for huge knowledge coverage, contributing to its efficacy in broad applications. In the context of CoT reasoning, Chen et al. (2023) conjuncture that LLM can be exposed to certain CoTs during training and easily complete reasoning by knowledge recall. As such, it is natural to adapt the basic but prevalent query augmentation technique in the term-based retrieval domain Dai & Callan (2019), which repeats the original query multiple times over the augmented part Wang et al. (2023b); Shen et al. (2023), into prompting LLMs.

## 3 METHODOLOGY

### 3.1 VANILLA CHAIN-OF-THOUGHT FOR REASONING

We begin with a unified formulation to leverage LLMs with CoT prompting to solve multiple natural language processing (NLP) and natural language understanding (NLU) tasks. In formal, given an input $x$ and target $y$, a LLM (parameterized as $\theta$) with CoT can be formulated as

$$y \sim \sum_{z \sim P(z \mid c^{(\text{cot})}(x); \theta)} P(\mathrm{y} \mid c^{(\text{cot})}(x, z); \theta) \cdot P(z \mid c^{(\text{cot})}(x); \theta), \tag{1}$$

where $c^{(\text{cot})}(\cdot)$ denotes a prompt template involving CoT-specific instructions like '*let's think step by step*', z stands for a latent variable of rationale and $z$ denotes a sampled rationale in natural language. As such, the LLMs could break down complex tasks into more manageable reasoning steps, treating each step as a piece of the overall solution chain. We take CoT as a baseline to solve reasoning tasks without sacrificing its generality. More extensively, our proposed simple RE2 can serve as a "plug & play" module to most other prompting methods (§3.3).

### 3.2 RE-READING (RE2) IMPROVES REASONING

To achieve a "bidirectional" comprehension within the context of unidirectional LLMs and to place a greater emphasis on processing input, we propose the Re-reading (RE2) strategy. Re-reading emerges as a fundamental strategy in human cognition when faced with intricate questions or statements. Especially in complex reasoning scenarios, individuals tend to revisit the information sources, be it a text or a diagram, to reassess the contex refine understanding, and correct potential misconceptions. Analogously, for LLMs to effectively tackle such complex tasks, implementing a re-reading strategy can be advantageous.

Intuitively, our proposed RE2 strategy encapsulates a dual-pass mechanism, where the first pass scans the input context in its entirety, and the subsequent re-read pass emphasizes refining understanding by focusing on salient regions. With RE2, the Eq. 1 can be readily rephrased as:

$$y \sim \sum_{z \sim P(z \mid c^{(\text{cot})}(\text{re2}(x)); \theta)} P(\mathrm{y} \mid c^{(\text{cot})}(\text{re2}(x), z); \theta) \cdot P(z \mid c^{(\text{cot})}(\text{re2}(x)); \theta), \tag{2}$$

where $\text{re2}(\cdot)$ is the re-reading operation of the input. We don't seek complex adjustments or intricate computational overhead for LLMs but a general implementation of $\text{re2}(x)$ simplicity as follows:

```
Q: {Input Query}
Read the question again: {Input Query}                              (3)
#Thought-eliciting prompt (e.g.,''Let's think step by step")#
```

where '{Input Query}' is a placeholder for the core target query, $x$. As such, the RE2 strategy attempts to emulate human-like revisitation of textual information to improve comprehension and reasoning in LLMs. Beyond, emphasizing the input through re-reading can enhance knowledge recall in a parametric retrieval manner. By reintroducing the input query, the model can better align its response with pre-existing knowledge or patterns. It's analogous to how a human, upon revisiting a problem statement or question, might remember a similar problem they've solved before.

## 3.3 GENERALITY OF RE2

The true power of the RE2 lies in its universality, offering adaptability across a range of tasks without necessitating significant architectural modifications. At its core, RE2 taps into the primary cognitive mechanisms by which humans process information, promoting depth of understanding through iterative engagement with textual data. This is particularly salient in the world of language models, where context reigns supreme. Hence, this general approach can be seamlessly integrated into various models and algorithms, such as few-shot setting, self-consistency, various thought-eliciting prompting strategies, and so on. We provide detailed insight into the integration of RE2 with other thought-eliciting prompting strategies as an illustration.

**Compatibility with Thought-Eliciting Prompting Strategies.** The prevailing body of research in this field has primarily emphasized the development of diverse thought-eliciting prompting. These prompting strategies guide and channel reasoning processes in generating output. In contras RE2 shifts its focus towards input, engaging in multiple passes over the provided information, thereby enhancing its comprehension of the question at hand. Consequently, RE2 exhibits a fair compatibility with respect to these thought-eliciting prompting strategies and can seamlessly serve as a 'plug & play' module alongside them. This synergy holds the potential to further enhance the reasoning abilities of LLMs.

With a specific strategy $s$ for eliciting thoughts from the LLMs, like Chain-of-Thoughts Wei et al. (2022b), Plan-and-Solve Wang et al. (2023a), Program-Aided Prompt Gao et al. (2023), and so on, the Eq. (2) is rewritten as:

$$y \sim \sum_{z \sim P(\text{z}|\,\text{c}^{(s)}(\text{re2}(x));\theta)} P(\text{y}|\,\text{c}^{(s)}(\text{re2}(x), z); \theta) \cdot P(z|\,\text{c}^{(s)}(\text{re2}(x)); \theta), \qquad (4)$$

## 4 EXPERIMENTS

We carried out a set of experiments to confirm the efficacy of the proposed RE2 across various reasoning assessments. Our findings indicate that across a wide range of tasks, models, and prompting methods, re-reading generally enhances the performance of reasoning in language models.

## 4.1 BENCHMARKS

We assess the effect of RE2 prompting strategy across three key categories of reasoning benchmarks.

**Arithmetic Reasoning** We consider the following seven arithmetic reasoning benchmarks: the GSM8K benchmark of math word problems Cobbe et al. (2021), the SVAMP dataset of math word problems with varying structures Patel et al. (2021), the ASDiv dataset of diverse math word problems Miao et al. (2020), the AQuA dataset of algebraic word problems Ling et al. (2017), the AddSub Hosseini et al. (2014) of math word problems on addition and subtraction for third, fourth, and fifth grader, MultiArith Roy & Roth (2015) dataset of math problems with multiple steps, and the SingelEQ Roy et al. (2015) dataset of elementary math word problems with single operation.

Table 1: Evaluation results on arithmetic reasoning benchmarks.

| LLMs | Methods | GSM | SVAMP | ASDIV | AQUA | MultiArith | SinlgeEQ | AddSub |
|------|---------|-----|-------|-------|------|-----------|----------|--------|
| davinci-003 | Vanilla | 19.48 | 67.60 | 69.00 | 28.74 | 31.33 | 86.22 | 89.87 |
| | Vanilla+RE2 | **24.79** | **70.90** | **71.20** | **30.31** | **42.33** | **87.20** | **92.15** |
| | | ↑ 5.31 | ↑ 3.30 | ↑ 2.20 | ↑ 1.57 | ↑ 11.00 | ↑ 0.98 | ↑ 2.28 |
| | CoT | 58.98 | 78.30 | 77.60 | 40.55 | 89.33 | 92.32 | 91.39 |
| | CoT+RE2 | **61.64** | **81.00** | **78.60** | **44.49** | **93.33** | **93.31** | **91.65** |
| | | ↑ 2.68 | ↑ 2.70 | ↑ 1.00 | ↑ 3.94 | ↑ 4.00 | ↑ 0.99 | ↑ 0.26 |
| ChatGPT | Vanilla | 77.79 | 81.50 | 87.00 | **63.39** | **97.83** | **95.28** | **92.41** |
| | Vanilla+RE2 | **79.45** | **84.20** | **88.40** | 58.27 | 96.67 | 94.49 | 91.65 |
| | | ↑ 1.66 | ↑ 2.70 | ↑ 0.60 | ↓ 5.12 | ↓ 1.16 | ↓ 0.79 | ↓ 0.76 |
| | CoT | 78.77 | 78.70 | 85.60 | 55.91 | 95.50 | 93.70 | 88.61 |
| | CoT+RE2 | **80.59** | **80.00** | **86.00** | **59.06** | **96.50** | **95.28** | **89.87** |
| | | ↑ 1.82 | ↑ 1.30 | ↑ 0.40 | ↑ 3.15 | ↑ 1.00 | ↑ 1.58 | ↑ 1.26 |

**Commonsense and Symbolic Reasoning** For Commonsense reasoning tasks, we used CommonsenseQA Talmor et al. (2019), StrategyQA Geva et al. (2021), and the AI2 Reasoning Challenge (ARC) Clark et al. (2018). CommonsenseQA dataset consists of questions that necessitate various common-sense knowledge. The StrategyQA benchmark dataset comprises questions that demand multi-step reasoning. The ARC dataset (denoted as ARC-t) is divided into two sets: a Challenge Set (denoted as ARC-c), containing questions that both retrieval-based and word co-occurrence algorithms answered incorrectly, and an Easy Set (denoted as ARC-e). We evaluate two symbolic reasoning tasks: date understanding Suzgun et al. (2023b) and Coinflip Wei et al. (2022b). Date understanding is a subset of BigBench datasets Suzgun et al. (2023b), which have posed challenges for previous fine-tuning efforts. Coinflip is a dataset of questions on whether a coin is still heads up after it is flipped or not flipped based on steps given in the questions.

## 4.2 LANGUAGE MODELS AND IMPLEMENTATIONS

In our implementation, we rigorously evaluate the performance of our RE2 model on two baseline prompting methods: Vanilla and CoT. The Vanilla approach aligns with the standard prompting method outlined in Wei et al. (2022b); Kojima et al. (2022), wherein no specific prompts are employed to elicit thoughts from the Language Models (LLMs). Conversely, the CoT method guides the model through a step-by-step thought process. We incorporate our RE2 strategy into these baseline methods to assess its impact, denoted as Vanilla+RE2 and CoT+RE2. To avoid the impact of randomness introduced by the demonstrations in a few-shot setting, we mainly assess our method in a zero-shot setting, following Chen et al. (2023); Wang et al. (2023a); Du et al. (2023). Additionally, for different tasks, we design answer-format instructions in prompts to regulate the structure of the final answer, facilitating precise answer extraction. Detailed information regarding the method prompts and answer-format instructions can be found in the paper's Appendix A.1. Moreover, we investigate the effectiveness of employing the re-reading mechanism in conjunction with various thought-eliciting prompting strategies, task settings, pretrained LLMs and self-consistency, as detailed in Section 4.4 of this paper. Our decoding strategy involves using greedy decoding with a temperature setting of 0, as well as self-consistency prompting with a temperature setting of 0.7. For these experiments, we employ two powerful backbones: ChatGPT (gpt-3.5-turbo-0613) OpenAI (2022) and davinci-003 (text-davinci-003)[1] , across all prompting methods, including Vanilla, CoT, Vanilla+RE2, and CoT+RE2.

## 4.3 EVALUATION RESULTS

Table 1 presents a comprehensive performance comparison between our method and existing zero-shot techniques on arithmetic reasoning datasets. Our analysis reveals a consistent enhancement in arithmetic reasoning attributed to re-reading, clearly outperforming both chain-of-thought prompting and vanilla prompting on almost all benchmarks when employing the davinci-003 model.

---

[1]https://platform.openai.com/docs/models/gpt-3-5

Table 2: Evaluation results on commonsense and symbolic reasoning benchmarks.

| LLMs | Methods | Commonsense | | | | | Symbolic | |
|------|---------|-------------|---|---|---|---|----------|---|
| | | CommonsenseQA | StrategyQA | ARC-e | ARC-c | ARC-t | Date | Coin |
| davinci-003 | Vanilla | 74.20 | 59.74 | 84.81 | 72.01 | 80.58 | 40.92 | 49.80 |
| | Vanilla+RE2 | **76.99** | **59.91** | **88.22** | **75.68** | **84.07** | **42.01** | **52.40** |
| | | ↑ 2.79 | ↑ 0.17 | ↑ 3.41 | ↑ 3.67 | ↑ 3.49 | ↑ 1.09 | ↑ 2.60 |
| | CoT | 71.66 | **67.55** | 85.69 | 73.21 | 81.57 | 46.07 | 95.60 |
| | CoT+RE2 | **73.05** | 66.24 | **87.84** | **76.02** | **83.94** | **52.57** | **99.60** |
| | | ↑ 1.39 | ↓ 1.31 | ↑ 2.15 | ↑ 2.81 | ↑ 2.37 | ↑ 6.50 | ↑ 4.00 |
| ChatGPT | Vanilla | 76.66 | 62.36 | **94.32** | **85.41** | **91.37** | 47.43 | 52.00 |
| | Vanilla+RE2 | **78.38** | **66.99** | 93.81 | 83.19 | 90.30 | **47.97** | **57.20** |
| | | ↑ 1.72 | ↑ 4.63 | ↓ 0.51 | ↓ 2.22 | ↓ 1.07 | ↑ 0.54 | ↑ 5.20 |
| | CoT | 69.94 | 67.82 | **93.35** | 83.53 | 90.11 | 43.63 | 88.80 |
| | CoT+RE2 | **71.66** | **69.34** | 93.14 | **84.47** | **90.27** | **47.15** | **95.20** |
| | | ↑ 1.72 | ↑ 1.52 | ↓ 0.21 | ↑ 0.94 | ↑ 0.16 | ↑ 3.52 | ↑ 6.40 |

Furthermore, when applied to ChatGPT, re-reading exhibits a substantial improvement in arithmetic reasoning performance on most datasets when combined with chain-of-thought prompting. For the vanilla prompting strategy, however, our method results in a performance drop on several benchmarks, including AQUA, MultiArith, SinlgeEQ, and AddSub.

Without clear instruction (i.e., "let's think step by step") for the chain-of-through mindset as in the CoT-prompting strategy, some general chat-based LLMs (e.g., ChatGPT, Claude) will likely keep performing chain-reasoning towards a final answer instead of writing the answer directly. The kinds of reasoning chains depend heavily on certain mindsets existing in alignment data, including repetition of user task instruction anthropic (2023). The case in anthropic (2023) proves that some existing LLMs have been trained to retell or paraphrase users' instructions to enhance the query-understanding capability, sharing a high-level inspiration with our method but leading to high learning costs to acquire this capability. As such, an overlay between the instruction-retell mindset and our re-reading strategy leads to more frequent repetition of users' instructions. As analyzed in the first part of §4.4 (i.e., *Times of Question Reading*) and empirically verified in Table 3, repeating the question multiple times results in worse results, thus closely aligned and consistent with the experimental results here in Table 1. From another perspective, initial findings outlined in Chen et al. (2023) suggest that during the instruction fine-tuning (IFT), ChatGPT was exposed to training samples containing CoT explanation, so the vanilla ChatGPT is prone to internalizing some of the CoTs and recalling the CoTs even without specific instructions. Chen et al. (2023) found that explicit CoT instructions sometimes yield worse results compared to using vanilla prompts with ChatGPT Chen et al. (2023) As such, introducing our re-reading prompt may not align closely with the CoT recalling mechanism by more attention paid to the query itself, potentially causing distraction to the implicit CoT instruction. Although davinci-003 has also undergone IFT training, it's worth noting that the generated outputs of the vanilla davinci–003 tend to lack CoT explanations. In situations where explanations are absent, understanding the problem becomes even more crucial. Consequently, the adoption of the re-reading strategy has shown great potential in enhancing performance in this scenario.

Table 2 presents the evaluation results for both commonsense reasoning and symbolic reasoning. We can discern a generally consistent performance trend mirroring that of the arithmetic reasoning tasks. Notably, our re-reading approach exhibits enhanced robustness and substantial improvement, particularly on davinci-003 and ChatGPT with CoT method. Additionally, our attention analysis demonstrates that more attention can be directed towards the weight and "bidirectional" comprehension of the question (see Appendix B).

## 4.4 DISCUSSIONS

**Times of Question Reading** We delve deeper into the impact of the times of question re-reading on reasoning performance. Table 3 illustrates how the performance of two distinct language models evolves concerning various times of question re-reading. An overarching pattern emerges across all models: performance improves until the number of re-reads reaches 2 or 3, after which it begins to decline with further increases in question re-reading times. The potential reasons for inferior performance when reading the question multiple times are two-fold: i) repeating the question in

Table 3: Evaluation results of the times of re-reading on GSM benchmark.

| **ChatGPT** | | | | | |
|---|---|---|---|---|---|
| | m=1 | m=2 | m=3 | m=4 | m=5 |
| Vanilla | 77.79 | **79.45** | 79.15 | 78.77 | 77.56 |
| CoT | 78.77 | 80.59 | **80.89** | 80.29 | 80.29 |
| **davinci-003** | | | | | |
| | m=1 | m=2 | m=3 | m=4 | m=5 |
| Vanilla | 19.48 | **24.79** | 18.80 | 18.35 | 17.06 |
| CoT | 58.98 | **61.64** | 60.12 | 58.83 | 57.47 |

Table 4: Evaluation results of some contemporary prompting strategies with re-reading.

| LLMs | Methods | GSM |
|---|---|---|
| ChatGPT | PS | 75.59 |
| | PS+RE2 | **76.27** |
| | PAL | 75.59 |
| | PAL + RE2 | **79.38** |
| davinci-003 | PS | 55.65 |
| | PS+RE2 | **58.68** |
| | PAL | 68.61 |
| | PAL + RE2 | **70.20** |

Table 5: Evaluation results on arithmetic reasoning benchmarks under few-shot setting.

| LLMs | Methods | GSM | SVAMP | ASDIV | AQUA | MultiArith | SinlgeEQ | AddSub |
|---|---|---|---|---|---|---|---|---|
| davinci-003 | Vanilla | 16.98 | 69.10 | 70.56 | **28.34** | 38.67 | 83.46 | 88.86 |
| | Vanilla+RE2 | **19.02** | **73.60** | **73.23** | 27.95 | **46.00** | **84.06** | **89.37** |
| | | ↑ 2.04 | ↑ 4.50 | ↑ 2.67 | ↓ 0.39 | ↑ 7.33 | ↑ 0.60 | ↑ 0.51 |
| | CoT | 56.63 | 78.90 | 79.96 | **46.45** | 96.16 | 90.94 | 88.60 |
| | CoT+RE2 | **60.12** | **79.80** | **81.21** | 44.89 | **96.83** | **91.14** | **89.37** |
| | | ↑ 3.49 | ↑ 0.90 | ↑ 1.25 | ↓ 1.56 | ↑ 0.67 | ↑ 0.20 | ↑ 0.77 |

a brute-force manner may interfere with the self-attention mechanism behind the LLMs, leading to over-weighed attention paid to the question only, and ii) repeating the question significantly increase the inconsistency of the LLMs between our inference and pretraining/alignment (intuitively in the learning corpora, we usually repeat a question twice to emphasize the key part, rather not more). It's noteworthy that reading the question two times tends to be optimal for accommodating most scenarios in our experiments, which is why we refer to this practice as "re-reading" in our paper.

**Compatibility with Thought-Eliciting Prompt Strategies** Compared to previous methods attempting to elicit thoughts in the output from LLMs, our RE2 emphasizes the understanding of the input. Therefore, we are intrigued to explore whether RE2 is effective with various thought-eliciting prompting strategies, aside from CoT. To investigate this, we apply RE2 to two other recently introduced prompting methods, namely, Plan-and-Solve (PS) Wang et al. (2023a) and Program-Aided Language models (PAL) Gao et al. (2023). The former model devises a plan to divide the entire task into smaller subtasks, and then carries out the subtasks according to the plan, while the latter generates programs as the intermediate reasoning steps. We directly apply our RE2 to these two methods by making a simple alteration to the input, following the prompt in Equation 3. Table 4 presents the evaluation findings on the GSM benchmark. Our observations reveal a consistent trend, akin to what was observed with chain-of-thought prompting. These results suggest that the effectiveness of our RE2 mechanism generally extends across various prompting methodologies.

**Compatibility with Few-Shot Prompting** It is noteworthy that our proposed re-reading mechanism is compatible with few-shot prompting. To demonstrate this compatibility, we conducted experiments on arithmetic reasoning tasks using the davinci-003 model, employing both Vanilla and CoT prompting methods. The few-shot prompting strategy and exemplars used align with those presented in Wei et al. (2022b). For both the Vanilla+RE2 and CoT+RE2 methods, we applied the re-reading mechanism to the exemplars as well. The results of these experiments are presented in Table 5. We can observe that the inclusion of the re-reading mechanism consistently enhances the performance of both prompting methods, mirroring our findings in the zero-shot setting.

**Effect on Non-IFT Models** In our primary experiments, we employed the ChatGPT and davinci-003 models, which had undergone IFT training. These models, being aligned with human-like behavior, are better equipped to follow instructions effectively. Additionally, they may have been exposed to datasets with CoT prompting during their training, making the "re-reading" mechanism potentially more beneficial in recalling explanations. To gauge the broader applicability of our approach and to eliminate any IFT-related impacts, we conducted experiments on non-IFT pretrained models: Llama-2-13B and Llama-2-70B Touvron et al. (2023b). Llama-2 is an open-source model pretrained on publicly available data without IFT or RLHF fine-tuning. We evaluated Llama-2 on

Table 6: Evaluation results of LLAMA-2 on arithmetic reasoning benchmarks

| LLMs | Methods | GSM | SVAMP | ASDIV | AQUA | MultiArith | SinlgeEQ | AddSub |
|------|---------|-----|-------|-------|------|-----------|----------|--------|
| Llama-2-13B | Vanilla | 5.76 | 43.90 | 52.91 | **22.44** | 6.33 | 68.11 | 66.58 |
| | Vanilla+RE2 | **6.82** | **47.90** | **53.15** | 17.32 | **6.50** | **69.68** | **70.12** |
| | | ↑ 1.06 | ↑ 4.00 | ↑ 0.24 | ↓ 5.12 | ↑ 0.17 | ↑ 1.57 | ↑ 3.54 |
| | CoT | 21.99 | 41.60 | 45.18 | 22.83 | **56.83** | 58.46 | 58.99 |
| | CoT+RE2 | **22.37** | **46.50** | **48.81** | **24.80** | 55.83 | **66.34** | **60.76** |
| | | ↑ 0.38 | ↑ 4.90 | ↑ 3.63 | ↑ 1.97 | ↓ 0.99 | ↑ 7.88 | ↑ 1.77 |
| Llama-2-70B | Vanilla | 11.60 | 56.60 | 61.31 | 20.08 | 24.67 | 77.17 | 80.25 |
| | Vanilla+RE2 | **13.50** | **63.60** | **64.66** | **22.05** | **25.00** | **80.31** | **84.05** |
| | | ↑ 1.90 | ↑ 7.00 | ↑ 3.35 | ↑ 1.97 | ↑ 0.33 | ↑ 3.14 | ↑ 3.80 |
| | CoT | 49.73 | 66.90 | 68.08 | 37.80 | 79.83 | 80.51 | 74.18 |
| | CoT+RE2 | **56.71** | **70.40** | **70.42** | **38.58** | **88.83** | **81.10** | **69.37** |
| | | ↑ 6.98 | ↑ 3.50 | ↑ 2.34 | ↑ 0.78 | ↑ 9.00 | ↑ 0.59 | ↓ 4.81 |

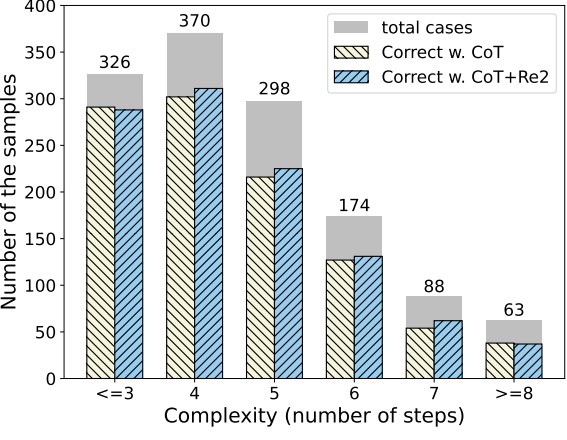

Figure 2: Model performance versus complexity of questions. X-axis means the complexity of questions and Y-axis refers to frequency. The gray hist means the number of total cases for each complexity.

Figure 3: N-gram recall between the generation and the input question. We take the question and generation as the reference and hypothesis respectively.

arithmetic reasoning tasks under a zero-shot setting, following Kojima et al. (2022). The results are presented in Table 6. The results clearly indicate that the re-reading mechanism consistently enhances the performance of both Vanilla and CoT prompting methods across most tasks when applied to Llama-2 models. This observation underscores the generality of our approach and dispels concerns about potential data leakage from IFT during training. This also underscores the versatility of RE2, which can be effectively employed across various model scales and types, regardless of whether they have undergone IFT training or are non-IFT models.

**Compatibility with Self-consistency**  Existing research indicates that the chain-of-thought prompting approach can be enhanced by adopting the self-consistency method, which involves aggregating the majority final answer from multiple sampled generations. We are also intrigued by the potential for further enhancing the proposed re-reading mechanism using this method. Consequently, we conducted experiments testing the integration of re-reading with the self-consistency approach on the GSM benchmark, and the results are presented in Table 7. Our findings demonstrate that self-consistency significantly enhances the performance of both prompting methods. Despite self-consistency's aggregation of multiple answers, our re-reading mechanism still contributes to improvement on most scenarios, indicating its compatibility with the self-consistency approach.

Table 7: Evaluation results of re-reading with self-consistency on GSM benchmark.

| LLMs | Methods | GSM | SVAMP |
|------|---------|-----|-------|
| ChatGPT | Vanilla | 77.79 | 81.50 |
| | Vanilla+SC | **85.52** | 87.50 |
| | Vanilla+RE2 +SC | 85.22 | **87.70** |
| | CoT | 78.77 | 78.70 |
| | CoT+SC | 85.75 | 84.90 |
| | CoT+RE2 +SC | **86.88** | **87.70** |

**Performance across Different Question Complexity.** We further investigate the impact of input question complexity on the reasoning performance of both CoT and CoT+RE2. In accordance with Fu et al. (2022), we measure question complexity by counting the reasoning steps present in the ground-truth explanations. Figure 2 illustrates how these models' performance evolves in response to varying question complexities. Our findings reveal a noticeable trend: the performance of all models generally diminishes as question complexity increases, suggesting that the current models still struggle with handling intricate queries. Notably, while employing a re-reading strategy leads to a slight drop in performance on less complex questions ($\leq 3$), the introduction of re-reading significantly enhances performance on more complex questions (e.g., those with a complexity level exceeding 5). This observation underscores the benefits of employing a re-reading strategy for improving question comprehension and reasoning capabilities over more complex questions. To further validate the improved understanding ability, we calculate the coverage degree (n-gram recall) between the generations and the input questions, as illustrated in Figure 3. The results indicate that RE2 improves the n-gram (n=1,2,3,4) recall in the output explanations, underscoring how our method enhances the model's focus on the question during the reasoning process to a certain extent.

**The impact of different re-reading instructions** We further conduct experiments to examine the influence of re-reading within the context of CoT prompting. Specifically, we initiate the investigation by comparing various instructions for question re-reading. As depicted in P1 and P2 in Table 8, instruction P1, which includes the phrase "Read the question again:", exhibits superior performance compared to directly repeating the question twice (referred to as P0). These results suggest that providing more detailed re-reading instructions to the language models is advantageous. Subsequently, we explore the possibility of introducing re-reading for CoT instructions (referred to as "Let's think step by step"), as exemplified in P3 and P4. However, we observe that repeating the thinking process two times does not yield any discernible benefits. Since this aspect is not the primary focus of this paper, we have deferred it to future research endeavors. It's noteworthy that, in general, question re-reading consistently improves reasoning performance compared to the standard CoT prompting without question re-reading (P0).

Table 8: Results of different re-reading instructions.

| Prompt | | Vanilla | CoT |
|---|---|---|---|
| P0 | Q: {question}
#Answer format instruction#
A: Let's think step by step. | 77.79 | 78.77 |
| P1 | Q: {question}
Read the question again: {question}
#Answer format instruction#
A: Let's think step by step. | **79.45** | **80.59** |
| P2 | Q: {question}
Q: {question}
#Answer format instruction#
A: Let's think step by step. | 78.09 | 79.38 |
| P3 | Q: {question}
A: Let's think step by step.
Read the question again: {question}
#Answer format instruction#
A: Let's think step by step. | 79.08 | 80.36 |
| P4 | Q: {question}
A: Let's think step by step.
Q: {question}
#Answer format instruction#
A: Let's think step by step. | 78.09 | 79.38 |

## 5 CONCLUSION AND FUTURE WORKS

In this paper, we have delved into the concept of RE2 prompting, specifically focusing on "re-reading" question. This method stands out as a straightforward and widely applicable approach to enhance the reasoning capabilities of language models. Notably, RE2 aids in fostering bidirectional comprehension of questions within the context of decoder-only causal language models. Crucially, it operates independently of other thought-eliciting prompting strategies and ensemble techniques. Our comprehensive experiments encompassed a wide array of tasks, diverse model types, varied task configurations, and compatibility assessments with other prompting methods. These experiments validated both the efficacy and versatility of our RE2. Notably, it exhibits enhanced effectiveness when combined with thought-eliciting prompting strategies, such as CoT. Our findings encourage the research community to focus on a deeper understanding of input questions, complementing the exploration of thought-eliciting prompting strategies. While our RE2 method showcases commendable performance across a diverse spectrum of textual tasks, our future endeavors aim to further expand its versatility into additional domains. This includes exploring into computer vision and multi-modal tasks applications.

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

# A APPENDIX

## A.1 SPECIFIC PROMPTING METHODS

Table 9: Specific prompts of Vanilla, Vanilla+RE2, CoT, CoT+RE2, PS, and PS+RE2.

| Methods | Prompt Content |
|---|---|
| Vanilla | Q: {question}
#Answer format instruction#
A: |
| Vanilla+RE2 | Q: {question}
Read the question again: {question}
#Answer format instruction#
A: |
| CoT | Q: {question}
#Answer format instruction#
A: Let's think step by step. |
| CoT+RE2 | Q: {question}
Read the question again: {question}
#Answer format instruction#
A: Let's think step by step. |
| PS | Q: {question}
#Answer format instruction#
A: Let's first understand the problem and devise a plan to solve the problem.
Then, let's carry out the plan, solve the problem step by step,
and give the ultimate answer. Please explicitly generate the mentioned process:
[Problem Understanding], [Plan], [Solving/Calculations], [Answer]. in your response. |
| PS+RE2 | Q: {question}
Read the question again: {question}
#Answer format instruction#
A: Let's first understand the problem and devise a plan to solve the problem.
Then, let's carry out the plan, solve the problem step by step,
and give the ultimate answer. Please explicitly generate the mentioned process:
[Problem Understanding], [Plan], [Solving/Calculations], [Answer]. in your response. |

Table 10: Specific prompts of PAL and PAL+RE2

| Methods | Prompt Content |
|---------|----------------|
| PAL | ```#!/bin/python3
import math
import numpy as np
import statistics
import sympy as sp
########### Task Instruction ##############
# You will write python program to solve math problems.
# You will only write code blocks.
# Please generate your code block in 'def solution()' function, and thus it can be
executed by python interpreter. You don't need to call 'solution()' function because
it will be called by the system.
# The concrete format of 'solution()' is as follows:
# def solution():
# """"<question>"""
# <your code>
# result = <your result>
# return result
#########################################
# Q: {question}
# Your defined "solution()" function with comments here.``` |
| PAL+RE2 | ```#!/bin/python3
import math
import numpy as np
import statistics
import sympy as sp
########### Task Instruction ##############
# You will write python program to solve math problems.
# You will only write code blocks.
# Please generate your code block in 'def solution()' function, and thus it can be
executed by python interpreter. You don't need to call 'solution()' function because
it will be called by the system.
# The concrete format of 'solution()' is as follows:
# def solution():
# """"<question>"""
# <your code>
# result = <your result>
# return result
#########################################
# Q: {question}
# Read the question again: {question}
# Your defined "solution()" function with comments here.``` |

Table 11: Specific answer-format instruction in each task.

| Tasks | Answer-format Instructions |
|-------|---------------------------|
| GSM | Your final answer should be a single numerical number, in the form $\boxed{answer}$, at the end of your response. |
| SVAMP | Your final answer should be a single numerical number, in the form $\boxed{answer}$, at the end of your response. |
| ASDIV | Your final answer should be a single numerical number, in the form $\boxed{answer}$, at the end of your response. |
| AQUA | Your answer should be in the form $\boxed{choice}$. There is only one correct choice. |
| MultiArith | Your final answer should be a single numerical number, in the form $\boxed{answer}$, at the end of your response. |
| SingleEQ | Your final answer should be a single numerical number, in the form $\boxed{answer}$, at the end of your response. |
| AddSub | Your final answer should be a single numerical number, in the form $\boxed{answer}$, at the end of your response. |
| CommonsenseQA | Your answer should be in the form $\boxed{choice}$. There is only one correct choice. |
| StrategyQA | Your answer should be either $\boxed{yes}$ or $\boxed{no}$, in the form $\boxed{answer}$. |
| ARC | Your answer should be in the form $\boxed{choice}$. There is only one correct choice. |
| Date Understanding | Your answer should be a date, in the format of $\boxed{MM/DD/YYYY}$, e.g. $\boxed{05/01/2022}$. |
| Coin Flip | Your answer should be either $\boxed{yes}$ or $\boxed{no}$, in the form $\boxed{answer}$. |

# B ATTENTION ANALYSIS

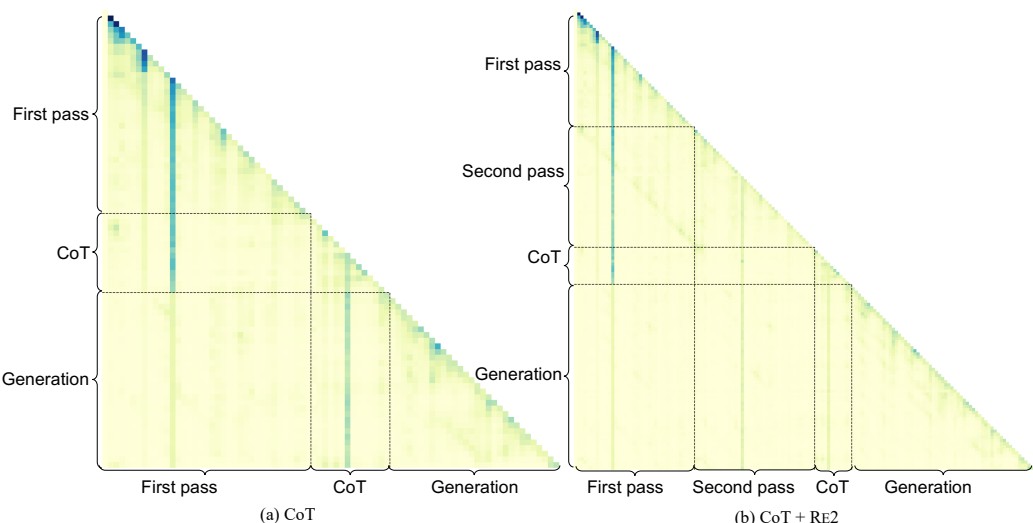

Figure 4: Attention visualization with and without RE2. (a) CoT prompting: there is only one pass for the question. (b) CoT+RE2 re-reads the question, including first pass and second pass. The row of matrix represents the query tokens and the column represents the key tokens.

To gain deeper insights into how RE2 reshapes attention during inference, we visualize the attention distribution by computing the average attention weights across all heads and layers in Llama-2. The results are presented in Figure 4, revealing two key findings: (1) In the block of "Second pass" attending to the "First pass" as shown in (b) for CoT+RE2, we observe explicit attentions in the upper triangle. This observation indicates that tokens in the second question can focus on the tokens behind the corresponding positions in the first question. In this way, RE2 enables a "bidirectional" understanding of the question. Notably, with the inclusion of RE2, the generation process maintains a higher attention weight on the question tokens. By calculating the proportion of attention weights assigned to the question tokens during generation, we observe an increase from 0.32 to 0.40 with the utilization of RE2. This finding suggests that the re-reading mechanism enhances the model's focus on the question during the reasoning process.

# C CASE STUDY

We also conduct a case study to show the effectiveness of our proposed re-reading prompting over the chain-of-thought. We choose two examples from GSM, and the results are listed in Table 12-13. It is evident that our method can better align the evidence in the question with the corresponding explanation hints. We can observe that CoT+RE2 tends to highlight the important evidences in the question before generating the explanation, for example, "*In the morning, she gives 15 cups of feed, and in the afternoon, she gives another 25. So ...*" in Table 12 and "*The bonus is worth half a month's salary, which is ...*" in Table 13. This observation is also consistent with Figure 3. Tables 14-18 provide more examples.

Table 12: GSM Example 4 (Complexity=3)

**Q:** Every day, Wendi feeds each of her chickens three cups of mixed chicken feed, containing seeds, mealworms and vegetables to help keep them healthy. She gives the chickens their feed in three separate meals. In the morning, she gives her flock of chickens 15 cups of feed. In the afternoon, she gives her chickens another 25 cups of feed. How many cups of feed does she need to give her chickens in the final meal of the day if the size of Wendi's flock is 20 chickens? (Answer: 20)

**CoT**

**Input:**
Q: Every day, Wendi feeds each of her chickens three cups of mixed chicken feed, containing seeds, mealworms and vegetables to help keep them healthy. She gives the chickens their feed in three separate meals. In the morning, she gives her flock of chickens 15 cups of feed. In the afternoon, she gives her chickens another 25 cups of feed. How many cups of feed does she need to give her chickens in the final meal of the day if the size of Wendi's flock is 20 chickens?
Your final answer should be a single numerical number, in the form \boxed{answer}, at the end of your response.
A: Let's think step by step.

**Output:**
First, let's calculate how much feed Wendi gives her chickens in the morning. She has 20 chickens, and she gives each chicken 3 cups of feed, so in total, she gives them 20 x 3 = 60 cups of feed in the morning.
Next, let's calculate how much feed Wendi gives her chickens in the afternoon. Again, she has 20 chickens, and she gives each chicken 3 cups of feed, so in total, she gives them 20 x 3 = 60 cups of feed in the afternoon.
To find out how much feed she needs to give her chickens in the final meal of the day, we need to subtract the total feed given in the morning and afternoon from the total feed needed for the day. The total feed needed for the day for 20 chickens is 20 x 3 x 3 = 180 cups of feed.
So, the feed needed in the final meal of the day is 180 - 60 - 60 = 60 cups of feed.
Therefore, Wendi needs to give her chickens 60 cups of feed in the final meal of the day.
Answer: $\boxed{60}$

**CoT+RE2**

**Input:**
Q: Every day, Wendi feeds each of her chickens three cups of mixed chicken feed, containing seeds, mealworms and vegetables to help keep them healthy. She gives the chickens their feed in three separate meals. In the morning, she gives her flock of chickens 15 cups of feed. In the afternoon, she gives her chickens another 25 cups of feed. How many cups of feed does she need to give her chickens in the final meal of the day if the size of Wendi's flock is 20 chickens?
Read the question again: Every day, Wendi feeds each of her chickens three cups of mixed chicken feed, containing seeds, mealworms and vegetables to help keep them healthy. She gives the chickens their feed in three separate meals. In the morning, she gives her flock of chickens 15 cups of feed. In the afternoon, she gives her chickens another 25 cups of feed. How many cups of feed does she need to give her chickens in the final meal of the day if the size of Wendi's flock is 20 chickens?
A: Let's think step by step.

**Output:**
First, we need to determine the total amount of feed Wendi gives her chickens in the morning and afternoon. In the morning, she gives 15 cups of feed, and in the afternoon, she gives another 25 cups. So, the total amount of feed given in the morning and afternoon is 15 + 25 = 40 cups.
Next, we need to determine the total amount of feed needed for Wendi's flock of 20 chickens. Since each chicken requires 3 cups of feed, we multiply 3 by 20 to get 60 cups.
Finally, we subtract the total amount of feed given in the morning and afternoon from the total amount needed. 60 - 40 = 20
Therefore, Wendi needs to give her chickens 20 cups of feed in the final meal of the day.
Answer: $\boxed{20}$

Table 13: GSM Example 641 (Complexity=6)

**Q:** Tim gets a promotion that offers him a 5% raise on his $20000 a month salary. It also gives him a bonus worth half a month's salary. How much money will he make in a year? (Answer: 262500)

### CoT

**Input:**
Q: Tim gets a promotion that offers him a 5% raise on his $20000 a month salary. It also gives him a bonus worth half a month's salary. How much money will he make in a year?
Your final answer should be a single numerical number, in the form \boxed{answer}, at the end of your response.
A: Let's think step by step.
**Output:**
First, let's calculate the 5% raise on Tim's $20000 monthly salary. To do this, we multiply $20000 by 5% (or 0.05):
$20000 * 0.05 = $1000
So, Tim's raise is $1000 per month.
Next, let's calculate the bonus worth half a month's salary. Half of $20000 is $10000, so Tim's bonus is $10000.
Now, let's calculate Tim's total monthly salary after the raise and bonus:
$20000 (original salary) + $1000 (raise) + $10000 (bonus) = $31000
Finally, let's calculate Tim's annual salary by multiplying his monthly salary by 12:
$31000 * 12 = $372000
Therefore, Tim will make $372000 in a year. So, the answer is $\boxed{372000}$.

### CoT+RE2

**Input:**
Q: Tim gets a promotion that offers him a 5% raise on his $20000 a month salary. It also gives him a bonus worth half a month's salary. How much money will he make in a year?
Read the question again: Tim gets a promotion that offers him a 5% raise on his $20000 a month salary. It also gives him a bonus worth half a month's salary. How much money will he make in a year?
Your final answer should be a single numerical number, in the form \boxed{answer}, at the end of your response.
A: Let's think step by step.
**Output:**
First, let's calculate the 5% raise on Tim's $20000 monthly salary.
The raise amount would be 5% of $20000, which is equal to 0.05 * $20000 = $1000.
So, Tim's new monthly salary would be $20000 + $1000 = $21000.
Next, let's calculate the bonus amount. The bonus is worth half a month's salary, which is equal to 0.5 * $21000 = $10500.
Now, to find out how much money Tim will make in a year, we need to calculate the total salary for 12 months and add the bonus.
Total salary for 12 months = $21000 * 12 = $252000.
So, Tim will make a total of $252000 + $10500 = $262500 in a year.
Therefore, Tim will make $\boxed{262500}$ in a year.

## C.1 MORE CASES

Table 14: GSM Example 926 (Complexity¿=9)

---

**Q:** Colby loves going to the movies and every month his parents give him $150 to spend at the movies. Tickets for Fridays and Saturdays cost $10. Tickets for any other day cost $7. Popcorn costs $8 and boxes of candy cost $2. It is the last day of the month and it's a Friday. He wants to make sure he gets a popcorn and box of candy that night. How many movies can he see if he already saw 5 movies on a Friday or Saturday, 8 movies on other days, had 2 tubs of popcorn, and four boxes of candy that month? (Answer: 1)

---

**CoT**

**Input:**

Q: Colby loves going to the movies and every month his parents give him $150 to spend at the movies. Tickets for Fridays and Saturdays cost $10. Tickets for any other day cost $7. Popcorn costs $8 and boxes of candy cost $2. It is the last day of the month and it's a Friday. He wants to make sure he gets a popcorn and box of candy that night. How many movies can he see if he already saw 5 movies on a Friday or Saturday, 8 movies on other days, had 2 tubs of popcorn, and four boxes of candy that month?

Your final answer should be a single numerical number, in the form \boxed{answer}, at the end of your response.

A: Let's think step by step.

**Output:**

First, let's calculate the cost of the movies Colby has already seen. He has seen 5 movies on a Friday or Saturday, which cost $10 each, so that is a total of 5 * $10 = $50. He has also seen 8 movies on other days, which cost $7 each, so that is a total of 8 * $7 = $56. Therefore, the total cost of the movies he has already seen is $50 + $56 = $106.

Next, let's calculate the cost of the popcorn and candy he wants to buy on the last day of the month. The popcorn costs $8 and the candy costs $2, so the total cost of the snacks is $8 + $2 = $10.

Now, let's subtract the cost of the movies Colby has already seen and the cost of the snacks from the $150 his parents gave him. $150 - $106 - $10 = $34.

Since each movie on a Friday or Saturday costs $10 and Colby has $34 left, he can see a maximum of $34 / $10 = 3 movies on a Friday or Saturday.

In conclusion, Colby can see a total of 3 + 5 = 8 movies in the month.

8

---

**CoT+RE2**

**Input:**

Q: Colby loves going to the movies and every month his parents give him $150 to spend at the movies. Tickets for Fridays and Saturdays cost $10. Tickets for any other day cost $7. Popcorn costs $8 and boxes of candy cost $2. It is the last day of the month and it's a Friday. He wants to make sure he gets a popcorn and box of candy that night. How many movies can he see if he already saw 5 movies on a Friday or Saturday, 8 movies on other days, had 2 tubs of popcorn, and four boxes of candy that month?

Read the question again: Colby loves going to the movies and every month his parents give him $150 to spend at the movies. Tickets for Fridays and Saturdays cost $10. Tickets for any other day cost $7. Popcorn costs $8 and boxes of candy cost $2. It is the last day of the month and it's a Friday. He wants to make sure he gets a popcorn and box of candy that night. How many movies can he see if he already saw 5 movies on a Friday or Saturday, 8 movies on other days, had 2 tubs of popcorn, and four boxes of candy that month?

A: Let's think step by step.

**Output:**

First, let's calculate the total amount Colby has spent so far:

For the 5 movies on Fridays and Saturdays, he spent 5 x $10 = $50. For the 8 movies on other days, he spent 8 x $7 = $56. For the 2 tubs of popcorn, he spent 2 x $8 = $16. For the 4 boxes of candy, he spent 4 x $2 = $8.

Adding all these expenses together, Colby has spent $50 + $56 + $16 + $8 = $130.

Now, let's calculate how much money he has left from the $150 his parents gave him:

Colby started with $150 - $130 = $20.

Since it's the last day of the month and it's a Friday, he wants to buy a popcorn ($8) and a box of candy ($2). This means he will spend $8 + $2 = $10.

Therefore, Colby will have $20 - $10 = $10 left.

Now, let's calculate how many movies he can see with the remaining $10:

Since tickets on Fridays and Saturdays cost $10, and he has $10 left, he can see 1 movie on a Friday or Saturday.

Therefore, Colby can see 1 more movie on the last day of the month.

So, the final answer is 1.

---

Table 15: GSM Example 242 (Complexity=6)

**Q:** Mike was a pen pal with 5 people. He stopped being penpals with 2 of them. They each send 2 letters a week that are 5 pages long. He responds in kind. He can write a page every 6 minutes. How many hours does he spend writing a week? (Answer: 3)

### CoT

**Input:**

Q: Mike was a pen pal with 5 people. He stopped being penpals with 2 of them. They each send 2 letters a week that are 5 pages long. He responds in kind. He can write a page every 6 minutes. How many hours does he spend writing a week?

Your final answer should be a single numerical number, in the form \boxed{answer}, at the end of your response.

A: Let's think step by step.

**Output:**

First, let's calculate how many letters Mike receives in a week from the 5 pen pals. Each pen pal sends 2 letters per week, so 5 pen pals would send him 5 * 2 = 10 letters in a week.

Next, let's calculate how many pages Mike needs to write in response to each letter. Each letter is 5 pages long, and Mike responds in kind, so he would write 5 pages for each letter he receives.

Therefore, Mike needs to write a total of 10 * 5 = 50 pages in a week.

Now, let's calculate how many minutes it takes for Mike to write one page. We are given that he can write a page in 6 minutes.

Since Mike needs to write 50 pages, it will take him 50 * 6 = 300 minutes to complete all the writing.

Lastly, let's convert the minutes to hours. There are 60 minutes in an hour, so 300 minutes is equal to 300 / 60 = 5 hours.

Therefore, Mike spends 5 hours writing in a week.

Answer: $\boxed{5}$.

### CoT+RE2

**Input:**

Q: Mike was a pen pal with 5 people. He stopped being penpals with 2 of them. They each send 2 letters a week that are 5 pages long. He responds in kind. He can write a page every 6 minutes. How many hours does he spend writing a week?

Read the question again: Mike was a pen pal with 5 people. He stopped being penpals with 2 of them. They each send 2 letters a week that are 5 pages long. He responds in kind. He can write a page every 6 minutes. How many hours does he spend writing a week?

A: Let's think step by step.

**Output:**

First, let's calculate the total number of letters that Mike receives in a week. Since he was pen pals with 5 people, and he stopped being pen pals with 2 of them, he is still receiving letters from 5 - 2 = 3 people.

Each of these 3 people sends 2 letters a week, so the total number of letters that Mike receives in a week is 3 * 2 = 6 letters.

Next, let's calculate the total number of pages that Mike needs to write in response. Each letter is 5 pages long, so the total number of pages that Mike needs to write is 6 * 5 = 30 pages.

Since Mike can write a page every 6 minutes, the total time he spends writing in a week is 30 * 6 = 180 minutes.

To convert this to hours, we divide by 60 since there are 60 minutes in an hour. So, Mike spends 180 / 60 = 3 hours writing in a week.

Therefore, Mike spends $\boxed{3}$ hours writing a week.

Table 16: GSM Example 273 (Complexity=6)

**Q:** Terri is knitting a sweater with two sleeves, a collar, and a decorative rosette. The body of the sweater takes 900 stitches to complete, the collar takes a tenth of that number of stitches, and the rosette takes twice as many as the collar. The whole sweater is an 1800-stitch project. How many stitches does each sleeve take? (Answer: 315)

### CoT

**Input:**
Q: Terri is knitting a sweater with two sleeves, a collar, and a decorative rosette. The body of the sweater takes 900 stitches to complete, the collar takes a tenth of that number of stitches, and the rosette takes twice as many as the collar. The whole sweater is an 1800-stitch project. How many stitches does each sleeve take?
Your final answer should be a single numerical number, in the form \boxed{answer}, at the end of your response.
A: Let's think step by step.

**Output:**
First, we know that the body of the sweater takes 900 stitches to complete.
Next, we know that the collar takes a tenth of the number of stitches as the body. So, the collar takes 900 stitches / 10 = 90 stitches.
Finally, we know that the rosette takes twice as many stitches as the collar. So, the rosette takes 90 stitches * 2 = 180 stitches.
To find out how many stitches each sleeve takes, we need to subtract the stitches of the body, collar, and rosette from the total number of stitches in the sweater.
So, each sleeve takes 1800 stitches - 900 stitches (body) - 90 stitches (collar) - 180 stitches (rosette) = 630 stitches.
Therefore, each sleeve takes ⁠630⁠ stitches.

### CoT+Re2

**Input:**
Q: Terri is knitting a sweater with two sleeves, a collar, and a decorative rosette. The body of the sweater takes 900 stitches to complete, the collar takes a tenth of that number of stitches, and the rosette takes twice as many as the collar. The whole sweater is an 1800-stitch project. How many stitches does each sleeve take?
Read the question again: Terri is knitting a sweater with two sleeves, a collar, and a decorative rosette. The body of the sweater takes 900 stitches to complete, the collar takes a tenth of that number of stitches, and the rosette takes twice as many as the collar. The whole sweater is an 1800-stitch project. How many stitches does each sleeve take?
A: Let's think step by step.

**Output:**
First, we know that the body of the sweater takes 900 stitches to complete.
Next, we are told that the collar takes a tenth of the number of stitches in the body of the sweater. Since the body has 900 stitches, the collar will have 900 / 10 = 90 stitches.
Finally, we are told that the rosette takes twice as many stitches as the collar. Since the collar has 90 stitches, the rosette will have 90 * 2 = 180 stitches.
Now, let's add up all the stitches to find the total number of stitches in the sweater: 900 (body) + 90 (collar) + 180 (rosette) = 1170 stitches.
We are given that the whole sweater is an 1800-stitch project. So, the remaining stitches must be for the sleeves.
1800 (total stitches) - 1170 (stitches accounted for) = 630 stitches for the sleeves.
Since there are two sleeves, each sleeve will have 630 / 2 = 315 stitches.
Therefore, each sleeve takes 315 stitches.

Table 17: GSM Example 101 (Complexity=6)

---

**Q:** Solo has to read 4 pages from his Science textbook, 20 pages from his Social Studies textbook, 7 pages from his History textbook and 8 pages from his Geography textbook. Solo read 15 pages on Monday. If he has 4 more days to complete all his reading, how many pages does he need to read, on average, in one day? (Answer: 6)

---

### CoT

**Input:**

Q: Solo has to read 4 pages from his Science textbook, 20 pages from his Social Studies textbook, 7 pages from his History textbook and 8 pages from his Geography textbook. Solo read 15 pages on Monday. If he has 4 more days to complete all his reading, how many pages does he need to read, on average, in one day?

Your final answer should be a single numerical number, in the form \boxed{answer}, at the end of your response.

A: Let's think step by step.

**Output:**

First, let's calculate how many pages Solo has left to read. He needs to read 4 pages from his Science textbook, 20 pages from his Social Studies textbook, 7 pages from his History textbook, and 8 pages from his Geography textbook.

So, the total number of pages he needs to read is 4 + 20 + 7 + 8 = 39 pages.

Since Solo has 4 more days to complete all his reading, we can divide the total number of pages by the number of days.

Therefore, Solo needs to read an average of 39 pages / 4 days = 9.75 pages per day.

Therefore, Solo needs to read an average of $\boxed{9.75}$ pages per day.

---

### CoT+RE2

**Input:**

Q: Solo has to read 4 pages from his Science textbook, 20 pages from his Social Studies textbook, 7 pages from his History textbook and 8 pages from his Geography textbook. Solo read 15 pages on Monday. If he has 4 more days to complete all his reading, how many pages does he need to read, on average, in one day?

Read the question again: Solo has to read 4 pages from his Science textbook, 20 pages from his Social Studies textbook, 7 pages from his History textbook and 8 pages from his Geography textbook. Solo read 15 pages on Monday. If he has 4 more days to complete all his reading, how many pages does he need to read, on average, in one day?

A: Let's think step by step.

**Output:**

First, let's calculate the total number of pages Solo needs to read. He needs to read 4 pages from his Science textbook, 20 pages from his Social Studies textbook, 7 pages from his History textbook, and 8 pages from his Geography textbook. So, the total number of pages he needs to read is 4 + 20 + 7 + 8 = 39 pages.

Next, let's calculate the number of pages he has left to read after Monday. He read 15 pages on Monday, so he has 39 - 15 = 24 pages left to read.

Finally, let's calculate the average number of pages he needs to read per day. He has 4 more days to complete his reading, so he needs to read 24 / 4 = 6 pages per day.

Therefore, Solo needs to read an average of $\boxed{6}$ pages per day.

---

Table 18: StrategyQA Example 262

---

**Q:** Would a nickel fit inside a koala pouch? (Answer: yes)

---

**CoT**

**Input:**
Q: Would a nickel fit inside a koala pouch?
Your answer should be either \boxed{yes} or \boxed{no}, in the form \boxed{answer}.
A: Let's think step by step.
**Output:**
First, let's consider the size of a nickel. A nickel has a diameter of approximately 21.21 mm and a thickness of approximately 1.95 mm.
Next, let's consider the size of a koala pouch. A koala pouch is designed to hold a baby koala, also known as a joey. The size of a koala pouch can vary, but it is generally not large enough to fit objects the size of a nickel.
Therefore, based on the size of a nickel and the size of a koala pouch, it is unlikely that a nickel would fit inside a koala pouch.
$\boxed{No}$

---

**CoT+RE2**

**Input:**
Q: Would a nickel fit inside a koala pouch?
Read the question again: Would a nickel fit inside a koala pouch?
Your answer should be either \boxed{yes} or \boxed{no}, in the form \boxed{answer}.
A: Let's think step by step.
**Output:**
First, let's consider the size of a nickel. A nickel has a diameter of about 0.835 inches (21.21 mm) and a thickness of about 0.077 inches (1.95 mm).
Next, let's consider the size of a koala pouch. A koala pouch is designed to hold a baby koala, also known as a joey. The pouch is stretchy and expands as the joey grows. However, the size of the pouch may vary depending on the age and size of the joey.
Based on the size of a nickel and the stretchiness of a koala pouch, it is possible that a nickel could fit inside a koala pouch. However, it would depend on the size of the pouch and the position of the joey inside.
So, the answer is $\boxed{yes}$, a nickel could potentially fit inside a koala pouch.

---

