# OpenReview forum: "Re-Reading Improves Reasoning in Language Models"
_ICLR.cc/2024/Conference — Submitted to ICLR 2024_

### Official Review · Reviewer_ZszC · 2023-10-31

**Soundness:** 3 good
**Presentation:** 4 excellent
**Contribution:** 3 good
**Rating:** 8
**Confidence:** 5

**Summary:**

This paper introduces a simple prompting strategy RE2 which re-reads the question multiple times. The authors demonstrate the effectiveness of the RE2 on a set of reasoning benchmarks either in the vanilla setting or in combination with other techniques including CoT, PS, PAL and self-consistency. They also conducted ablation studies on the times of re-reading, complexity of questions, and different re-reading instructions.

**Strengths:**

1. The biggest strength of the paper to me is the simplicity of the method, making it easily adoptable by the wide research community.
2. The paper is very comprehensive in reasoning datasets covered, models evaluated on, baselines compared against and ablations conducted.
3. The results are mostly positive against the baselines for all the datasets and models studied.

**Weaknesses:**

1. The gains are more pronounced in weaker models (davinci-003 vs ChatGPT, Llama-2-13B vs 70B). This raises the question of the scaling behavior of the proposed RE2 method.
2. For ARC tasks evaluated on ChatGPT, RE2 shows negative or neutral-ish results in both the vanilla and CoT settings. This is concerning, and worth more investigations to understand why.
3. A lot of the gains in the paper are within the range of 2%, and it is unclear whether these results are just noise since the paper didn’t provide any way to quantify the standard deviations.

**Questions:**

1. Typo in the last sentence of the abstract: “though-eliciting prompting…”.
2. The claim of “LLMs to understand the input in a bidirectional manner” is misleading: it is unclear to me where the bidirectional attention from the model comes from. Neither did the authors explain what exactly they mean by “bidirectional”.
3. The authors claim that LLMs gain deeper insights/understanding with RE2. However this claim is not supported by any evidence at all. It can be totally misleading. For instance, re-reading 3 times is not better than 2 times. It is possible that in pretraining corpus, there is such data which resembles the re-reading 2 times behavior, giving an edge to RE2.
4. Ideally, RE2 should work beyond Reasoning given that humans don’t do re-reading on reasoning tasks. Covering tasks beyond Reasoning would certainly make the paper much stronger.

---

> ### Author Response · Authors · 2023-11-22
> **Response to Reviewer ZszC (1/2)**
>
> Dear Reviewer ZszC,
>
> Thank you for your review! In the following we would try to address your concerns.
>
> > W.1: "The gains are more pronounced in weaker models (davinci-003 vs ChatGPT, Llama-2-13B vs 70B). This raises the question of the scaling behavior of the proposed RE2 method."
>
> - Thank you for your insightful comments! We will try our best to respond to your concerns from two aspects.
> - (1) **davinci-003 vs. ChatGPT**: Many details of training differences, such as data, model size, and training methods, for ChatGPT and text-davinci-003 **have not been officially disclosed**. Additionally, it's important to note that ChatGPT and davinci-003 belong to **different model categories**, with ChatGPT being more focused on **dialogue scenarios**. Therefore, directly comparing their strengths and weaknesses can be challenging. Furthermore, regarding the model scale, ChatGPT appears to have **a smaller model size (20B)** compared to 175B davinci-003, as "leaked” in related Microsoft publications [1]. Therefore, to gain insights into the scaling behavior, we recommend considering open-source models for a more direct comparison.
>
> - (2) **Llama-13B vs. Llama-70B**: To investigate the scaling behavior on open-source LLMs, we calculate the average improvement percentages based on Table 6, and the results are presented in the table below
>
>     | Model | Average Gain |
>     |----------|----------|
>     | Llama-13B | 1.93 |
>     | Llama-70B | **2.71**|
> - As shown in the table, the improvements on **Llama-70B are more pronounced, which aligns with the scaling behavior**. We sincerely hope that the clarification provided can address your concern.
>
>     - [1] Mukul Singh, José Cambronero, Sumit Gulwani, Vu Le, Carina Negreanu, Gust Verbruggen: CodeFusion: A Pre-trained Diffusion Model for Code Generation. CoRRabs/2310.17680 (2023)
>
>
>
>
> > W.2: For ARC tasks evaluated on ChatGPT, RE2 shows negative or neutral-ish results in both the vanilla and CoT settings. This is concerning, and worth more investigations to understand why.
>
> - Thanks for your valuable comments. As for ChatGPT, we have also noticed that very few experiments (e.g., AQUA, MultiArith, SinlgeEQ, AddSub, and ARC) do not show improvement and provided discussion in Section 4.3.
>
> - As indicated in [2], during the instruction fine-tuning (IFT), **ChatGPT may be exposed to tasks containing CoT explanations**. Particularly, on before mentioned datasets, “ChatGPT with Vanilla” (i.e. without explicit CoT instructions) could still produce CoT output, and **“ChatGPT with CoT” even performs worse than “ChatGPT with Vanilla”**, which is evidenced by [2] and our experiments. Therefore, other explicit instructions might disrupt the learned pattern in ChatGPT, leading to decreased performance. Nevertheless, our Re2 method still could **achieve improvements in 71% of the experiments on ChatGPT**. We hope the above clarification can resolve your concerns.
>
>     - [2] Jiuhai Chen, Lichang Chen, Heng Huang, Tianyi Zhou: When do you need Chain-of-Thought Prompting for ChatGPT? CoRRabs/2304.03262 (2023)
>
>
>
> > W.3: A lot of the gains in the paper are within the range of 2%, and it is unclear whether these results are just noise since the paper didn’t provide any way to quantify the standard deviations.
>
> - Thanks for your valuable comment! First of all, **in line with prior studies [3][4]**, all our tests (except for self-consistency) use a **greedy decoding strategy**, leading to deterministic outputs. As a result, the responses for the same prompt exhibit negligible standard deviation.
>
> - Regarding the range of gains, we provide more statistical results here. Statistically, davinci-003 with Vanilla+Re2 shows average improvements of **3.81, 2.51, and 1.85** in arithmetic, commonsense, and symbolic tasks respectively; davinci-003 with CoT + Re2 showed average improvements of **2.22, 1.23, and 5.25** in the same categories; Llama-2-70B with Vanilla + Re2 shows an average improvement of **3.70** in arithmetic tasks; and Llama-2-70B with CoT + Re2 shows an average improvement of **2.63** in arithmetic tasks. Furthermore, lots of additional experiments conducted further substantiate the effectiveness of Re2. In summary, the range of improvements aligns comparably with findings from related studies on LLM reasoning, such as [3][4]. We hope the above clarification can resolve your concerns.
>
>     - [3] Lei Wang, Wanyu Xu, Yihuai Lan, Zhiqiang Hu, Yunshi Lan, Roy Ka-Wei Lee, Ee-Peng Lim: Plan-and-Solve Prompting: Improving Zero-Shot Chain-of-Thought Reasoningby Large Language Models. ACL (1) 2023: 2609-2634
>     - [4] Luyu Gao, Aman Madaan, Shuyan Zhou, Uri Alon, Pengfei Liu, Yiming Yang, Jamie Callan, Graham Neubig: PAL: Program-aided Language Models. ICML 2023: 10764-10799

---

> ### Author Response · Authors · 2023-11-22
> **Response to Reviewer ZszC (2/2)**
>
> > Q.1: Typo in the last sentence of the abstract: “though-eliciting prompting…”.
>
> - Thank you for kind reminders, and we have corrected typos.
>
>
>
> > Q.2: The claim of “LLMs to understand the input in a bidirectional manner” is misleading: it is unclear to me where the bidirectional attention from the model comes from. Neither did the authors explain what exactly they mean by “bidirectional”.
>
> - Thank you for the valuable comments! The concept of "bidirectional" in the context of our Re2 strategy refers to the enhanced scope of token visibility during the processing stages. In a standard unidirectional decoder-only model, **tokens within a question cannot 'see' subsequent tokens.** However, with the implementation of Re2, every token **in the second pass has visibility of the complete first pass**, including its "subsequent tokens" in the initial question. This mechanism effectively simulates a bidirectional effect. Thanks again for your comment, and we will ensure to clarify this more clearly in our revised manuscript.
>
> > Q.3: The authors claim that LLMs gain deeper insights/understanding with RE2. However this claim is not supported by any evidence at all. It can be totally misleading. For instance, re-reading 3 times is not better than 2 times. It is possible that in pretraining corpus, there is such data which resembles the re-reading 2 times behavior, giving an edge to RE2
>
> - Thank you greatly for your constructive comments! From the perspective of a working mechanism, repeating questions could **allocate more computational resources to the input encoding stage, and achieve a "bidirectional" encoding effect**.  That enables LLMs to attend to the question itself more. We also conducted two analytical experiments to verify this clarification. The n-gram recall in Figure 3 indicates that Re2 helps LLMs generate more content related to the question, improving focus during the generation stage. Additionally, **attention visualization** in Appendix B shows that tokens in the second pass do concentrate on the subsequent tokens of the first pass, enabling a "bidirectional" understanding. To avoid any potential misunderstandings, we will subsequently revise relevant statements to be more rigorous.
>
> - We acknowledge the remark of “Re2 in pretraining corpus” greatly, which has also been discussed in the paragraph "Times of Question Reading". Considering all the above, the effectiveness of Re2 may be multifaceted, including both the understanding ability and the pretraining data. Thanks again for your thoughtful comment.
>
>
> > Q.4: Ideally, RE2 should work beyond Reasoning given that humans don’t do re-reading on reasoning tasks. Covering tasks beyond Reasoning would certainly make the paper much stronger.
>
> - Thank you for your constructive suggestion regarding the expansion of Re2.
>
> - Many existing studies on Cognitive Science reveal that humans do re-reading during human learning and problem-solving [5][6][7]. For instance, [5] states that re-reading is a valuable study tool, encourages more efficient processing when learning knowledge, and [7] finds that most students adopt the strategy of re-reading to remedy comprehension failures and remember important information. Besides, the concept of re-reading has been successfully applied to some traditional NLP tasks (e.g., sentiment analysis and discourse relation recognition) [8][9]. Motivated by these studies, we propose Re2 to enhance reasoning in LLMs. Importantly, our conducted experiments showcase the effectiveness of Re2 on the reasoning scenario, and the analytical experiments (e.g. token coverage and attention visualization) reveal that LLMs with Re2 could focus more on the question. We sincerely hope this clarification could address your concern.
>
> - Regarding the applied scenarios, we are also considering Re2 on richer domains, such as image-based question answering and document retrieval, which we consider to be highly promising and exciting directions. Thanks again for your great suggestion!
>
>     - [5] Dowhower, Sarah Lynn. "Effects of repeated reading on second-grade transitional readers' fluency and comprehension." Reading Research Quarterly (1987): 389-406.
>     - [6] Dowhower, Sarah L. "Repeated reading: Research into practice." The Reading Teacher 42.7 (1989): 502-507.
>     - [7] Ozek, Yesim, and Muharrem Civelek. "A study on the use of cognitive reading strategies by ELT students." The Asian EFL Journal 14.1 (2006): 1-26.
>     - [8] Lei Sha, Feng Qian, and Zhifang Sui. Will repeated reading benefit natural language understanding? NLPCC 2017
>     - [9] Yang Liu and Sujian Li. Recognizing implicit discourse relations via repeated reading: Neural networks with multi-level attention. EMNLP 2016

---

### Official Review · Reviewer_hwHe · 2023-11-04

**Soundness:** 2 fair
**Presentation:** 3 good
**Contribution:** 2 fair
**Rating:** 5
**Confidence:** 3

**Summary:**

This paper proposes RE2, a simple modification to improve the reasoning ability of large language models. As claimed in the paper, the existing decoder-only model can not well capture the back-and-forth interactions between different stages during reasoning. The authors simply repeat the question first before solving it. In this way, as claimed, earlier tokens can be aware of later tokens in the question. This approach are evaluated in several benchmarks including arithmetic, commonsense and symbolic. Many ablation studies are done to support the effectiveness of proposed RE2.

**Strengths:**

- To enable back-and-forth interaction during reasoning in large language models is a reasonable motivation, since a single-pass forward process in decoder-only architecture may not be sufficient for the complex reasoning process.

- The experiments are well designed and complementary, supporting the proposed repeated question prompts from several perspectives.

-  The paper is well organized and very easy to follow. I enjoy reading the paper.

**Weaknesses:**

- The authors connect the repeating question prompts with human's thinking process, which is a casual argument without justification to back this up. It is hard to be convinced this is how and why the repeated prompts help.

- Repeating the question needs a question assumed to be there. It seems not to be generalizable for many other scenarios where it is not simply a Q-A setting, such as a multi-round conversation. Instead, approaches like chain-of-thoughts are in the solving stage, i.e., they can be used in any scenario.

- In figure 2,  RE2 makes the low-complexity questions (<=3) worse in the GSM benchmark. However, the other arithmetic benchmarks (except GSM) in table1 5 6 are mostly in low complexity too. These two results are contradict. Why is this the case?

**Questions:**

- For most datasets in table1 and table2, it seems RE2 improves vanilla more than CoT, but the case is the opposite in the symbolic reasoning. Is there any interpretation of this difference?

- From table1 and table2, RE2 would almost always improve davinci-003 but seems pretty random in ChatGPT-vanilla (half better, half worse). Why do they behave in such a different way?

---

> ### Author Response · Authors · 2023-11-22
> **Response to Reviewer hwHe (1/2)**
>
> Dear Reviewer hwHe,
>
> Thank you for your detailed review! We will try to address your concerns.
>
> > W.1: "The authors connect the repeating question prompts with the human's thinking process, which is a causal argument without justification to back this up. It is hard to be convinced this is how and why the repeated prompts help."
>
> - We appreciate your constructive comments. What we intended to convey is that the Re2 strategy for LLMs is **inspired** by human cognitive processes, rather than aiming to strictly connect Re2 with human reading patterns. We agree that the connection between LLM's thinking and human cognition is an intricate and subtle matter, one that will require further exploration and research in the future. To avoid any potential misunderstandings, we will subsequently revise relevant statements to be more rigorous.
>
> - Regarding the working mechanism, our method aligns with information processing in neural networks, enhancing the model's depth of processing and allocating **more computing resources to encode input**. Additionally, the second pass of the question can see the complete input of the first pass, achieving a **"bidirectional" understanding within the unidirectional decoder-only LLM**. Importantly, we conducted experiments to analyze why Re2 is effective. Figure 3 shows that Re2 enhances the model's n-gram (n=1,2,3,4) recall in the output explanations, indicating that **Re2 improves the model's focus on the question during the generation stage**. Furthermore, in Appendix B, we also performed **attention visualization**. The results show that tokens in the second pass can focus on the tokens behind the corresponding positions in the first pass, thereby enabling a “bidirectional” understanding of the question through Re2. We hope the clarification provided may help address your concern.
>
>
>
> > W.2: Repeating the question needs a question assumed to be there. It seems not to be generalizable for many other scenarios where it is not simply a Q-A setting, such as a multi-round conversation.
>
> - Thanks for your great suggestion! Our work is primarily centered around improving LLMs’ reasoning capability, which is widely studied in NLP and LLM. **Consistent with many previous works (such as CoT [1] and Self-consistency [2])**, we conducted various experiments on three reasoning scenarios, including **arithmetic, commonsense and symbolic tasks, totaling 14 datasets.**
>
> - We are also exploring Re2 on richer domains, such as image-based question answering and document retrieval. Additionally, we deeply acknowledge that applying Re2 to multi-round conversations is an interesting and important scenario. However, unlike the single-round Q-A setting, it may require **more refined designs for the conversation scenario**, such as the selection of content and position to re-read. Nevertheless, we are very glad to investigate the impact of RE2 on this scenario in the future.
> - Thanks again for your great suggestion!
>
>     - [1] Jason Wei, Xuezhi Wang, Dale Schuurmans, Maarten Bosma, Brian Ichter, Fei Xia, Ed H. Chi, Quoc V. Le, Denny Zhou: Chain-of-Thought Prompting Elicits Reasoning in Large Language Models.NeurIPS 2022
>     - [2] Xuezhi Wang, Jason Wei, Dale Schuurmans, Quoc V. Le, Ed H. Chi, Sharan Narang, Aakanksha Chowdhery, Denny Z
>
>
>
> > W.3: "In figure 2, RE2 makes the low-complexity questions (<=3) worse in the GSM benchmark. However, the other arithmetic benchmarks (except GSM) in table1 5 6 are mostly in low complexity too. These two results are contradict. Why is this the case?"
>
> - We appreciate you bringing this to our attention, and we would like to resolve your concerns through more experimental analyses.
>
> - Due to the limited space, we **only present one setting regarding ChatGPT with CoT/CoT+Re2** for the complexity analysis. To establish generality, we conducted the same analysis on other settings, i.e. ChatGPT with Vanilla/Vanilla+Re2; davinci-003 + CoT/CoT+Re2; davinci-003 + Vanilla/Vanilla+Re2, specifically focusing on the subset of 326 samples with complexity <=3. The experimental results about the number of correct samples in this subset are shown as follows:
>
>
> ||  ChatGPT with CoT | ChatGPT with Vanilla | davinci-003 with CoT | davinci-003 with Vanilla |
> |----------|----------|----------|----------|----------|
> w/o Re2 | **291**|279|247|109|
> w/ Re2 | 288 | **287** | **259** | **131**|
>
> - The experimental results demonstrate that Re2 **consistently enhances the performance** of the baselines, which is consistent with the results observed in other datasets. Furthermore, for the ChatGPT with CoT+Re2 presented in Figure 2, the decrease in the number of correct samples is **very minimal, merely by 3 samples**. This can be considered a result that is within a comparable range. Thanks again for your thoughtful remark. We hope this clarification could address your concerns, and we will also include complete experiments to mitigate any further concerns.

---

> ### Author Response · Authors · 2023-11-22
> **Response to Reviewer hwHe (2/2)**
>
> > Q.1: For most datasets in table1 and table2, it seems RE2 improves vanilla more than CoT, but the case is the opposite in the symbolic reasoning. Is there any interpretation of this difference?
>
> - Thanks for your insightful comments! The feature of symbolic reasoning could bring insights to this phenomenon. Compared to other reasoning scenarios, **symbolic reasoning especially requires the model to infer a complete logical chain**. Therefore, symbolic reasoning relies more on the completeness of reasoning. Instead, other reasoning tasks primarily involve mathematical calculation or commonsense knowledge. We conducted further error analysis on 15 cases that “Davinci-003 with CoT” got incorrect in the Coin Flip dataset. The analysis shows that **93% errors could be categorized to “one step missing” type**, i.e. a step is missing in the logical chain. Notably, all of the cases with this error type are **fixed by “Davinci-003 with CoT+Re2”**, which aligns with the working mechanism of Re2 that LLMs with Re2 could focus on the question and the experiment result that Re2 could generate more tokens relevant to the question (cf. Figure 3). In contrast, **“one step missing” does not dominate in other scenarios**. For instance, [1] identifies that **other error types (e.g. calculation error)** are frequent occurrences in arithmetic reasoning, **accounting for 78% of errors in LLMs with CoT**. Therefore, Re2 could improve CoT more on symbolic reasoning by facilitating the complete logical chain. Thanks again for your constructive comments and hope that could resolve your concern :)
>
>
>     - [1] Jason Wei, Xuezhi Wang, Dale Schuurmans, Maarten Bosma, Brian Ichter, Fei Xia, Ed H. Chi, Quoc V. Le, Denny Zhou: Chain-of-Thought Prompting Elicits Reasoning in Large Language Models.NeurIPS 2022
>     - [3] Luyu Gao, Aman Madaan, Shuyan Zhou, Uri Alon, Pengfei Liu, Yiming Yang, Jamie Callan, Graham Neubig: PAL: Program-aided Language Models. ICML 2023: 10764-10799
>
>
> > Q.2: "From table1 and table2, RE2 would almost always improve davinci-003 but seems pretty random in ChatGPT-vanilla (half better, half worse). Why do they behave in such a different way?"
>
> - Thanks for your valuable comments. As for ChatGPT, we have also noticed that very few experiments (e.g., AQUA, MultiArith, SinlgeEQ, AddSub, and ARC) do not show improvement and provided discussion in Section 4.3. As indicated in [4], during the instruction fine-tuning (IFT), **ChatGPT may be exposed to tasks containing CoT explanations**. Particularly, on before mentioned datasets, “ChatGPT with Vanilla” (i.e. without explicit CoT instructions) could still produce CoT output, and **“ChatGPT with CoT” even performs worse than “ChatGPT with Vanilla”**, which is evidenced by [4] and our experiments. Therefore, other explicit instructions might disrupt the learned pattern in ChatGPT, leading to decreased performance. Nevertheless, our Re2 method still **could achieve improvements in 71% of the experiments on ChatGPT**. We hope the above clarification can resolve your concerns.
>
>     - [4] Jiuhai Chen, Lichang Chen, Heng Huang, Tianyi Zhou: When do you need Chain-of-Thought Prompting for ChatGPT? CoRRabs/2304.03262 (2023)

---

### Official Review · Reviewer_KNMq · 2023-11-07

**Soundness:** 3 good
**Presentation:** 4 excellent
**Contribution:** 2 fair
**Rating:** 5
**Confidence:** 3

**Summary:**

The paper proposed a simple yet interesting prompt, in which the question is repeated.  Experiments conducted on a series of reasoning benchmarks serve to underscore the effectiveness and generality of the proposed prompt.

**Strengths:**

The prompt proposed in the paper is interesting and simple enough, and demonstrated to be able to effectively improve the reasoning performance of LLMs. The presentation is clear and easy to follow.

**Weaknesses:**

The experiments conducted in the paper mainly compare the proposed method with vanilla COT with backbones ChatGPT and davinci-003 (Llama-2 is used for another reasoning task). But there have been lots of COT prompts recently, and other LLMs, which have not been evaluated in the paper. Even for the conducted experiments, the proposed method is not always useful for performance improvement, which can not fully support the theoretical analysis in the paper. To me, it's more suitable for a demonstration paper.

**Questions:**

In the experiment, why the backbone LLMs were divided into two groups for two sets of reasoning tasks, i.e. ChatGPT and davincci-003 for commonsense reasoning and symbolic reasoning, and Llama-2 for arithmetic reasoning? I think you should compare the competing methods with different LLMs in the same group of reasoniing tasks.

---

> ### Author Response · Authors · 2023-11-22
> **Response to Reviewer KNMq (1/2)**
>
> Dear Reviewer KNMq,
>
> Thank you for your review! We appreciate your feedback and will attempt to address your concerns.
>
> > W.1: "The experiments conducted in the paper mainly compare the proposed method with vanilla COT with backbones ChatGPT and davinci-003 (Llama-2 is used for another reasoning task). But there have been lots of COT prompts recently, and other LLMs, which have not been evaluated in the paper."
>
> - Thanks for your valuable questions! To demonstrate the versatility of our method, we have verified the effect of Re2 on other advanced CoT Prompts in **Section 4.4, "Compatibility with Thought-Eliciting Prompt Strategies"**. This includes two classic prompt methods: Plan-and-Solve (PS) [1] and Program-Aided Language (PAL) [2] methods. The former model devises a plan to break down the entire task into smaller subtasks and then executes them according to the plan. The latter generates programs as intermediate reasoning steps. Our experimental results show that these methods are effective; with davinci-003, the PS method's performance improved from 55.65 to 58.68, and the PAL method improved from 68.61 to 70.20 by applying Re2. The reviewer may kindly refer to Section 4.4 for more details.
>
> - Regarding the selection of LLMs, we have chosen **the most classic and commonly used LLMs in reasoning scenarios**, including ChatGPT, davinci-003, and Llama-2 (13B and 70B). On the one hand, these four models are **representative and have been widely used in existing reasoning studies [1][2][3]**. On the other hand, these models **represent two types of LLMs**. ChatGPT and davinci-003 are typical models of instruction fine-tuning (IFT), whereas Llama-2 is a representative public non-IFT pre-trained model.
>
> - We are glad to include more scenarios and LLMs for comparison if any suggestions are provided. Hope the above clarification can address your concern :)
>
>     - [1] Lei Wang, Wanyu Xu, Yihuai Lan, Zhiqiang Hu, Yunshi Lan, Roy Ka-Wei Lee, Ee-Peng Lim: Plan-and-Solve Prompting: Improving Zero-Shot Chain-of-Thought Reasoningby Large Language Models. ACL (1) 2023: 2609-2634
>     - [2] Luyu Gao, Aman Madaan, Shuyan Zhou, Uri Alon, Pengfei Liu, Yiming Yang, Jamie Callan, Graham Neubig: PAL: Program-aided Language Models. ICML 2023: 10764-10799
>     - [3] Ziwei Chai, Tianjie Zhang, Liang Wu, Kaiqiao Han, Xiaohai Hu, Xuanwen Huang, Yang Yang: GraphLLM: Boosting Graph Reasoning Ability of Large Language Model. CoRRabs/2310.05845 (2023)
>
>
> > W.2: "Even for the conducted experiments, the proposed method is not always useful for performance improvement, which can not fully support the theoretical analysis in the paper."
>
> - Thanks for your in-depth comments. Even though our method is not perfect on all 14 datasets with 112 settings, it does **show an overall improvement in almost all experiments**. Statistically, out of the 56 main experiments conducted on davinci-003 and Llama-2, 52 showed improvements, **accounting for 93% of the total**. The average accuracy improvement reaches **2.54%**. Specifically, davinci-003 with Vanilla+Re2 shows average improvements of **3.81, 2.51, and 1.85** in arithmetic, commonsense, and symbolic tasks respectively; davinci-003 with CoT + Re2 showed average improvements of **2.22, 1.23, and 5.25** in the same categories; Llama-2-70B with Vanilla + Re2 shows an average improvement of **3.70** in arithmetic tasks; and Llama-2-70B with CoT + Re2 shows an average improvement of **2.63** in arithmetic tasks. We humbly consider that these improvements demonstrate the versatility and effectiveness of our Re2 method.
>
> - As for ChatGPT, we also have discussed why very few experiments (e.g., AQUA, MultiArith, SinlgeEQ, AddSub, and ARC) do not show improvement. As indicated in [4], during the instruction fine-tuning (IFT), **ChatGPT may be exposed to tasks containing CoT explanations**. Particularly, on before mentioned datasets, “ChatGPT with Vanilla” (i.e. without explicit CoT instructions) could still produce CoT output, and **“ChatGPT with CoT” even performs worse than “ChatGPT with Vanilla”**, which is evidenced by [4]  and our experiments. Therefore, other explicit instructions might disrupt the learned pattern in ChatGPT, leading to decreased performance. Nevertheless, our Re2 method still could achieve improvements in **71% of the experiments** on ChatGPT. We hope the above clarification can resolve your concerns.
>
>     - [4] Jiuhai Chen, Lichang Chen, Heng Huang, Tianyi Zhou: When do you need Chain-of-Thought Prompting for ChatGPT? CoRRabs/2304.03262 (2023)

---

> ### Author Response · Authors · 2023-11-22
> **Response to Reviewer KNMq (2/2)**
>
> > W.3: "To me, it's more suitable for a demonstration paper."
>
> - We deeply appreciate your comment and would like to highlight the innovation and completeness of our research. We hope that the simplicity of our method does not lead to the misconception that it is more suited for a demonstration paper. The CoT research paper gained widespread attention precisely because of its **simplicity and effectiveness**. Most thought-eliciting prompting methods focus on guiding the model's thought process, allocating more computational resources to the decoding phase. In contrast, our Re2 is an original research finding that **allocates more computational resources to the encoding phase of the input, thus complementing most CoT prompts.** In addition, the second pass of the question can see the complete input of the first pass, **achieving a "bidirectional" understanding within the unidirectional decoder-only LLM**. Importantly, we conducted extensive experiments and analyses to verify our method, covering widely-used benchmarks, LLMs, various advanced reasoning prompts, zero-shot setting, few-shot setting, reading times, compatibility with self-consistency, variants of Re2 prompts and other analytical experiments. We humbly believe that our Re2 method provides a valuable perspective for LLM prompt design in the research area of LLM reasoning and may offer insights for future research in this field.
>
>
> > Q.1: In the experiment, why the backbone LLMs were divided into two groups for two sets of reasoning tasks, i.e. ChatGPT and davincci-003 for commonsense reasoning and symbolic reasoning, and Llama-2 for arithmetic reasoning? I think you should compare the competing methods with different LLMs in the same group of reasoning tasks.
>
> - Actually, both ChatGPT and davinci-003 also undergo arithmetic experiments, as indicated in Table 1, and the experiments of commonsense reasoning and symbolic reasoning are presented in Table 2.
> Furthermore, to carry on more analysis experiments, which include non-instruction fine-tuning models (Llama-2-13B and Llama-2-70B) and few-shot settings, we selected representative arithmetic tasks. We sincerely hope that the clarification could address your concern.

---

### Meta-Review · Area_Chair_wPSL · 2023-12-05

**Metareview:**

The paper presents a straightforward prompting strategy, Re2, which entails re-reading the question to enhance reasoning capabilities in Large Language Models (LLMs). This approach is commendable for its simplicity and its alignment with the cognitive principle of reinforcement, enabling bidirectional comprehension. The paper's strength lies in its comprehensive coverage of reasoning datasets, varied models, and extensive ablation studies, which collectively demonstrate the effectiveness of Re2 across different benchmarks and scenarios. However, the reviewers raised pertinent concerns regarding the method's scalability, particularly its varied effectiveness on different model sizes and tasks, and the lack of a robust theoretical justification linking the re-reading approach to improved reasoning performance. Additionally, the inconsistencies in performance improvement, particularly in low-complexity questions and certain tasks like the ARC, call for further investigation. The results, while positive, show marginal gains in some cases, raising questions about the statistical significance of the findings. Overall, the paper is currently a borderline paper, which contributes a potentially valuable approach to enhancing LLM reasoning, but would benefit from deeper theoretical analysis and broader evaluation to firmly establish its efficacy and applicability.

**Justification For Why Not Higher Score:**

While the technical solution is simple, but deeper theoretical analysis could be more useful.

**Justification For Why Not Lower Score:**

NA

---

### Decision · Program_Chairs · 2024-01-16

Reject